# Natural Product-Based Nanomedicine in Treatment of Inflammatory Bowel Disease

**DOI:** 10.3390/ijms21113956

**Published:** 2020-05-31

**Authors:** Tripti Khare, Sushesh Srivatsa Palakurthi, Brijesh M. Shah, Srinath Palakurthi, Sharad Khare

**Affiliations:** 1Division of Gastroenterology and Hepatology, Department of Medicine, University of Missouri, Columbia, MO 65212, USA; kharet@health.missouri.edu; 2Department of Pharmaceutical Sciences, Rangel College of Pharmacy, Texas A&M University, Kingsville, TX 78363, USA; susheshsrivatsa98@gmail.com (S.S.P.); brijeshshah@tamu.edu (B.M.S.); palakurthi@tamu.edu (S.P.); 3Harry S. Truman Veterans Hospital, Columbia, MO 65201, USA

**Keywords:** inflammatory bowel disease, Crohn’s disease, ulcerative colitis, phytochemicals, macromolecules, inflammation, nanoparticles, nanomedicine

## Abstract

Many synthetic drugs and monoclonal antibodies are currently in use to treat Inflammatory Bowel Disease (IBD). However, they all are implicated in causing severe side effects and long-term use results in many complications. Numerous in vitro and in vivo experiments demonstrate that phytochemicals and natural macromolecules from plants and animals reduce IBD-related complications with encouraging results. Additionally, many of them modify enzymatic activity, alleviate oxidative stress, and downregulate pro-inflammatory transcriptional factors and cytokine secretion. Translational significance of natural nanomedicine and strategies to investigate future natural product-based nanomedicine is discussed. Our focus in this review is to summarize the use of phytochemicals and macromolecules encapsulated in nanoparticles for the treatment of IBD and IBD-associated colorectal cancer.

## 1. Introduction

Inflammatory bowel disease (IBD), which includes Crohn’s disease (CD) and ulcerative colitis (UC), is a highly debilitating chronic inflammatory disorder of the small intestine and colon with enhanced risk of colorectal cancer (CRC). IBD affects millions of people and mainly occurs in genetically predisposed individuals having dysregulated immune response to various environmental conditions [1,2,3]. The burden of IBD is quite high, both on patient’s quality of life and on the health care system, with estimated hospitalization rates of 8.2–17 per 100,000 annually and annual treatment costs of $6.8 billion [4]. Genetic, immunological, and environmental factors play important roles in its etiology [5]. The most common symptoms include regular abdominal pain, fever, vomiting, diarrhea, blood in the stool, and weight loss with enhanced risk of colorectal cancer [6,7].

The factors involved in intestinal inflammation include altered synthesis and release of pro-inflammatory cytokines (interleukin (IL)-1β, IL-6, and IL-12), tumor necrosis factor α (TNF-α), interferon γ (IFN-γ), transforming growth factor (TGF)-β, and increased reactive oxygen species (ROS) that results in excessive damage to the intestinal tissues [6,8,9]. In IBD, intestinal permeability is predominantly abnormal with increased appearance of immune cells and increased mucus production [10,11,12]. The drug therapy in IBD principally induces either remission of acute attacks or limits the acute attacks during remission. Aminosalicylates, steroids, and immunosuppressants are some of the conventional anti-inflammatory drugs used for treating IBD. Although these medications are efficient, the systemic absorption of these anti-inflammatory medications results in side effects (short- and long-term), such as allergic responses, diarrhea, vomiting, lymphopenia, raised liver enzymes, and inflammation of the pancreas [1,13]. In addition to these conventional therapeutics, monoclonal antibody-based biological therapies are also recommended in IBD treatment. Several such USFDA (U.S. Food and Drug Administration) approved TNF-α antibodies include infliximab, adalizumab, certolizumab, vedolizumab, and golimumab, whereby infliximab was the first product approved for the management of IBD. However, huge costs associated to biological therapies, parenteral administration, higher number of nonresponders, and immune complications are major challenges limiting their potential in IBD treatment [14,15,16]. Therefore, curative strategies that may be more safe and effective in the treatment and management of IBD are required [5,6,17,18]

Several recent reports document use of natural phytochemicals in IBD that possess anti-inflammatory and antioxidant activity such as flavonoids and phenolic compounds. They modulate various inflammatory mediators, such as IL-1β, IL-6, IL-10, TNF-α, prostaglandin E2 (PGE-2), inducible nitric oxide synthase (iNOS), and cyclooxygenase-2 (COX-2) [13,19]. Additionally, natural macromolecules are also tested in suppression of IBD-associated biochemical and molecular inflammatory pathways. Recently, antioxidant and anti-inflammatory properties of biophenols derived from fruits and vegetables have been considered beneficial for IBD through various in vitro and in vivo studies [20]. Biophenols like oleuropein, hydroxytrosol, and ligstroside derived from olive trees have been reported to possess aforesaid activities [21]. In particular, oleuropein has been widely studied for its anti-inflammatory and immunomodulatory activities, whereby an oleuropein-treated colon biopsy sample of a colitis patient revealed restoration of the typical microscopic damage with strong recovery of mucin-forming goblet cells against untreated colon biopsy sample [22]. The same compound has also resulted in a reduction of pro-inflammatory cytokine level (IL-1β, IL-6, and IL-8) in ex-vivo organ cultures of mucosal explants from CD patients, thus demonstrating immunomodulatory activity in IBD [23]. Similarly, polyphenols derived from green tea have great therapeutic potential against IBD treatment due to their antioxidant properties, regulation of inflammatory mediator TNF-α, and COX-2 synthesis, thereby playing a vital role in downregulating the aberrant signaling pathways in IBD [24]. A possible barrier to this area is that the administration of natural drugs in conventional manner is restricted because of their little solubility, permeability, and bioavailability. However, micronization and nanonization of these natural drugs may be an acceptable strategy to improve their physical and chemical properties to overcome the challenges faced otherwise [25]. In the following pages, we will review encapsulated plant-based medicines and natural macromolecules in the treatment of colitis and colitis-associated colorectal cancer (CAC).

As use of natural products is an immensely vast field in IBD, we have focused this review on natural product-based nanomedicine in the treatment of IBD and associated CRC in preclinical models. Though use of phytochemicals and macromolecules have been discussed in IBD and IBD-CRC literature [26,27,28], the present review summarizes the studies which have used nanoparticles as a delivery platform. Also, we will not summarize pre- and probiotics, as this has been extensively reviewed by other investigators [29]. We performed a literature search in PubMed including the terms “IBD”, “colitis”, “natural products”, “phytochemical”, “herbal”, “macromolecule”, “colon”, and “inflammation” that resulted in around 900 references. We then sorted out the studies from this pool of references that have used “nanoparticles” as a delivery platform. To keep this review brief, we only focused on studies published during the last ten years.

## 2. Nanotechnology in IBD

Nanotechnology has recently emerged as promising future therapeutics for IBD and CAC. Lack of targeted delivery and inadequate drug availability at the select site predisposes many patients to the need for partial or total colectomy to prevent CAC. Nanoformulation of drugs overcomes the common barriers imposed by the colon such as a thick mucus layer, disrupted epithelium, and altered transit time [30]. Additionally, nano- and micro-sized drug delivery systems facilitate targeted delivery of medicines directly to the inflamed sites, increase their effective concentration at the sites, and reduce the side effects brought by systemic absorption of drugs [1,8,31]. Nanomedicine has several drug delivery systems, which include transferosomes; liposomes; dendrimers; mesoporous silica; solid lipids; microspheres; and cellular carriers like recombinant bacteria, macrophages, and erythrocytes [32]. These systems improve stability, specificity, bioavailability, and biodistribution of natural compounds [33]. Several nanosized systems proved effective in animal models of IBD, but approval for clinical use in humans is still scarce [34].

## 3. Natural Plant-Based Products as Nanomedicine in IBD Therapy

### 3.1. Thymoquinone

Thymoquinone (TQ) in the seed oil extract of *Nigella sativa* has several anti-inflammatory and immunomodulating activities targeting nuclear factor kappa B (NF-κB), IL-1β, and TNF-α signaling. Liposome or nanoparticle-based formulations of TQ are effective against various diseases in animal models [35]. Treatment of dextran sodium sulfate (DSS)-induced colitis in mice with TQ suppresses malondialdehyde (MDA) levels and myeloperoxidase (MPO) activity with concomitant increase in glutathione levels indicating improvement in colitis-associated tissue damage [36]. In addition, there is significant reduction in the expression of inflammatory markers Cox-2, iNOS, Nrf2, KEAP1, and pro-inflammatory cytokines (IL-1β, IL-6, and TNF-α) both at the mRNA and protein levels [37]. In vitro treatment of HT-29 colon cancer cells with a combination of TQ and lipopolysaccharide (LPS) also reduced inflammatory markers [38]. Further, oral administration of alginate microcapsule encapsulated *N. sativa* extract (NSE) is an efficient strategy for the delivery of TQ to the colon for the treatment of IBD [38].

### 3.2. Resveratrol

Resveratrol (RES) is a naturally occurring polyphenolic compound in red wine with antiplatelet, antitumor, neuroprotective, and anti-inflammatory properties. It regulates markers of inflammation by downregulating pro-inflammatory cytokines IL-1β, IL-6, and IL-8; TNF-α; and matrix metalloproteinase (MMP)-2, MMP-9, MMP-3, and MMP-13 in both in vivo and in vitro IBD models [39]. However, therapeutic use of RES is limited because of its rapid metabolism due to its little solubility in water and chemical instability. Iglesias et al. developed chitosan-based biocompatible hydrogels–nanoparticles (CTS–NPs) and used them as colon-specific drug delivery systems for the prolonged retention and release of resveratrol. Encapsulation of RES into CTS-NPs improves not only its absorption but also its distribution, metabolism, excretion, and toxicity [40].

### 3.3. Curcumin

Curcumin (Cur) is derived from the roots of a plant *Curcuma longa,* a member of the Zingiberaceae family. It is composed of 1,7-bis(4-hydroxy-3-methoxyphenyl)-1,6-heptadiene-3,5-dione, a polyphenolic hydrophobic compound, and 2%–5% turmeric, a bioactive pigment giving it a yellow color. Its anti-inflammatory [41,42], immunomodulatory [43,44,45], and antioxidant [46] properties are documented in several human diseases including cancers [47,48,49]. Traditional use of curcumin in the treatments is limited because of its poor absorption in the gastrointestinal tract, poor stability, low bioavailability, and rapid systemic elimination [50]. However, use of curcumin in nano-formulations with albumin, histone, solid lipids, polylactide-coglycolide, liposomes, and polybutylcyanocrylate improves its bioavailability, solubility, and stability, making it therapeutically stronger with no adverse effects [50,51].

Curcumin-primed and curcumin-encapsulated exosomes have shown profound anti-inflammatory activities by decreasing expression of IL-6 and TNF-α in murine macrophage RAW 264.7 cells when induced by lipopolysaccharide (LPS) [51]. Moreover, curcumin-primed and curcumin encapsulated exosomes are promising agents in treating inflammation-related diseases by affecting NF-κB-, Nrf2-, and activator of transcription-3 (STAT3)-dependent signaling pathways [52]. Qiao et al. characterized amphiphilic curcumin polymer (PCur), which is made of hydrophilic polyethylene glycol (PEG) and hydrophobic curcumin joined together by a disulfide bond. Due to nano-scaled sizes with sufficient solubility and neutral surface potential, PCur efficiently accumulates at the inflamed sites of the gut. Further, low cytotoxicity and increased membrane permeability of the PCur improves its oral bioavailability. Oral administration of PCur in DSS-induced mice results in amelioration of progression of inflammation in the colon and possible prevention from IBD and colitis-associated cancer (CAC) [53]. In another study, water-insoluble curcumin is chemically engineered into hydrophilic mucoadhesive chitosan and used in a preclinical dextran sodium sulfate (DSS) colitis model and azoxymethane (AOM)-DSS-induced CAC mouse models. Orally delivered curcumin-chitosan NPs accumulate in inflamed intestinal regions and tumor tissues. Treatment significantly protects mice from ulcerative colitis (UC) and CAC [54].

### 3.4. Ginger

The rhizome of *Zingiber officinale* is a medicinal plant, which is commonly known as ginger. Edible ginger-derived nanoparticles (GDNPs) have an average size of ~230 nm with negative zeta potential. The GDNPs are composed of few proteins, ~125 microRNAs, high levels of lipids, and large amounts of biologically active compounds (6-gingerol and 6-shogaol). GDNPs are shown to enhance intestinal repair and to reduce acute colitis and CAC in DSS and AOM-DSS mouse models, respectively. Increased survival and proliferation of intestinal epithelial cells (IECs), increased anti-inflammatory cytokines (IL-10 and IL-22), and reduced inflammatory cytokines (TNF-α, IL-6, and IL-1β) in response to oral GDNPs suggest its potential in decreasing damaging factors and in promoting healing effects [13]. Similarly, oral administration of siRNA-CD98/ginger-derived lipid vesicles (GDLVs) targets specifically to colon tissues, resulting in reduced expression of CD98 in colitis [7]. Plant-derived exosome-like nanoparticles (ELNs) that contain RNAs can alter microbiome composition and host physiology. In this regard, ginger ELNs (GELNs) ameliorate mouse colitis via IL-22-dependent mechanisms [55].

### 3.5. Quercetin

Quercetin (3,3′,4′,5,7-pentahydroxyflavone), a polyphenol found in abundance in onions, has anti-inflammatory and antioxidant activity [56]. It occurs commonly in both glycoside and aglycone forms [57]. PEG-coated vesicles with chitosan and nutriose-loaded quercetin are most suitable for the drug delivery in colon and thus improve and ameliorate symptoms of trinitrobenzene sulfonic acid (TNBS)-induced colitis in rats [58]. Glycoside-rutin is proven to be very effective in treating IBD in DSS-induced experimental animals by regulating body weight and oxidative stress in terms of MPO and by reducing GSH (glutathione), malondialdehyde, and serum nitrous oxide (NO) concentrations [59,60]. Lin et al. noticed that dietary quercetin alleviates the effects of *Citrobacter redentium*-induced colitis in mice by inhibiting the pro-inflammatory cytokines (IL-6, IL-17, and TNF-α) and by promoting anti-inflammatory cytokine IL-10 in colon tissues as well as by modifying gut microbiota [61]. Moreover, quercetin has bactericidal capacity and anti-inflammatory activity in macrophages via Heme Oxygenase-1 (HO-1)-mediated pathways and thus is useful in IBD therapy by restoring hemostasis and by balancing the enteric commensal microflora [62]. Another study by Dicarlo et al. revealed the benefit of quercetin in suppressing the inflammatory pathway in an in vitro intestinal organoid model. The authors stated that such a model is a novel tool to investigate epithelial response in models of chronic inflammation. Intestinal organoids of the Winnie model (ulcerative colitis mice model) treated with quercetin showed suppression of inflammation through downregulation of TNF-α and lipocalin-2 and upregulation of the heme oxygenase 1 and ferroportin 1 against the untreated model, thus mimicking the characteristics of the in vivo model in evaluating the gut epithelial inflammatory responses [63].

### 3.6. Embelin

Embelin (2,5-dihydroxy-3-undecyl-1,4-benzoquinone) is a naturally occurring alkyl substituted hydroxyl benzoquinone found abundantly in *Embelia ribes* plant. Embelin is used in traditional medicine for the treatment of various diseases [64]. Its fruit tastes bitter and is used commonly to treat fever, a variety of gastrointestinal ailments, and inflammatory diseases. Embelin possess anti-inflammatory, analgesic [65], antioxidant [66], and wound-healing activities [67] and is reported to inhibit TNF-α-induced activity of NF-κB, thereby impairing the inflammatory signaling. Standard particle size of 13.5 µm of embelin-loaded microspheres helps in delayed release at the inflamed sites and accumulation in macrophages. Embelin-loaded enteric-coated microspheres, which use both ethyl cellulose and Eudragit S 100 polymer, also exhibit delayed release as compared to plain embelin and exerts defensive effects in acetic acid-induced UC in rats by increasing GSH level and by reducing MPO and MDA levels [68]. Embelin also ameliorates DSS-induced colitis in mice by attenuating DAI (Disease Activity Index) scores and tissue MPO accumulation. It is also reported to prevent enlargement of spleen size and shortening of colon length in a dose-dependent manner. Additionally, Embelin exerts its anti-inflammatory activity by inhibiting secretion and abnormal expression of mRNA of pro-inflammatory cytokines (IL-1β, IL-6, and TNF-α) [69].

### 3.7. Grape Exosomes

In multicellular organisms, in addition to secretion of proteins, release of exosomes is another method of intercellular communication. Exosomes can haul proteins, lipids, mRNAs, and miRNAs from one cell to another and thus are involved in cell–cell communication. Recent research has shown that nanosized particles of plant origin can act as exosomes [70] and are involved in cell–cell communication, thereby regulating innate immunity [71]. The gastrointestinal tract is continuously in contact with digested edible plant derived exosome-like nanosized particles. However, the role of these plant-derived exosome-like nanoparticles as an interspecies messenger has never been addressed. Ju et al. identified exosome-like nanoparticles from grapes known as grape exosome-like nanoparticles (GELNPs) and demonstrated that GELNPs are 380.5 ± 37.47 nm in size and contain many miRNAs and proteins [72]. However, mammalian exosomes contain even greater numbers of miRNAs and proteins. The lipid profile of mammalian exosomes is also much different than that of GELNPs. GELNPs are composed of 2% galactolipids (typical plant lipids) and 98% phospholipids (50% of which is phosphatidic acid (PA)). Interaction of PA with the mammalian target of rapamycin (mTOR) is reported to initiate cell growth and proliferation [73]. Ju et al. also demonstrated that GELNPs can penetrate the intestinal mucus barrier and are involved in protection of colon against DSS-induced colitis in mice. In addition, lipids from GELNPs and Liposome-like nanoparticles (LLNs) are required for targeting nanoparticles towards intestinal stem cells. The signaling pathway mediated by β-catenin is blocked in GELNP recipient cells, which in turn modulates renewal and remodeling of intestinal tissues in response to pathological triggers [72].

### 3.8. Silymarin

Silymarin, isolated from *Silybum marianum* (milk thistle seeds), is a mixture of several flavanolignans (silybinin, silychristin, silydianin, and isosilyibinin) with antioxidant and anti-inflammatory properties. The activity of antioxidant enzymes such as glutathione peroxidase, superoxide dismutase, and catalase are affected by silymarin. In TNBS-induced murine colitis, silymarin, has been shown to rebalance inflammatory cytokines such as TNF-α, IL-1β, and IL-6. Similarly, silymarin and selenium NPs in combination significantly reduce expression of oxidative stress biomarkers, NF-κB, and pro-inflammatory cytokines and the drug combination was more effective than each one alone [74]. Varshosaz et al. on the other hand orally administered Eudragit NPs loaded with silybinin in acetic acid-induced UC animals and noticed a significant reduction in IL-6 and TNF-α activity, thus improving symptoms of IBD [75].

### 3.9. Caffeic Acid Phenethyl Ester

Honey bee propolis is found to contain a phenolic compound Caffeic acid phenethyl ester (CAPE) that is reported to possess anti-inflammatory activity mainly by inhibiting NF-κB [76]. In DSS-induced experimental mouse model of colitis, CAPE is effective in suppressing pro-inflammatory cytokines and MPO activity, which improves epithelial barrier function [77].

### 3.10. Piceatannol

Piceatannol (PCT) is a trans-3,4,3′,4′-tetrahydroxystilbene found in the seeds of *Euphorbia lagascae* and modulates activities of transcription factors NFκB, Nrf2, and HIF-1α [78,79,80]. Colon-targeted PCT regulates the production of transcription factors gene products in inflamed colonic tissues and thus increases the efficacy of PCT against colitis [81]. However, this activity is not observed in conventional PCT. Tambuwala et al. described the effect of albumin nano-encapsulation of CAPE and PCT on HIF-1α and nuclear p65, the important therapeutic targets of IBD in DSS-induced colitis. They inferred from their findings that nano-encapsulation of CAPE/PCT in albumin enhances its anti-inflammatory property and the ability to regulate molecular pathways related to inflammation [82].

## 4. Natural Macromolecules as Nanomedicine in IBD Therapy

### 4.1. Natural Peptides

Tuftsin, a natural tetrapeptide (Thr-Lys-Pro-Arg) with immunomodulating and anti-inflammatory activities, is a part of immunoglobulin G (IgG) heavy chain generated by enzymatic cleavage in the spleen. Several analogs of tuftsin are affective in colitis treatment in animal models [83]. Tuftsin-phosphocholine (TPC) is responsible for maintaining normal gut microbiota in collagen-induced arthritis model [84]. TPC immunomodulates by stimulating colon anti-inflammatory cytokines (IL-10) and by downregulating pro-inflammatory cytokines (IL-1β, IL-17, and TNF-α) in collagen-induced arthritis model and DSS-induced colitis model [85,86]. Lysine-proline-valine (KPV), a naturally occurring tripeptide, attenuates inflammatory responses of colonic cells. KPV entrapped into hyaluronic acid (HA)-functionalized polymeric nanoparticles and encapsulated in a hydrogel (chitosan/alginate) prevents mucosa damage and downregulates TNF-α [87].

### 4.2. Vasoactive Intestinal Peptide

Vasoactive Intestinal Peptide (VIP), an immunomodulating and anti-inflammatory endogenous hormone, reverses colitis and associated diarrhea in mouse models with sterically stabilized micelles (SSM) nanoformulation [88]. The receptor for VIP (VIPR1) is expressed differentially in UC and CD mucosa. It is expressed in CD3- and CD68-positive cells of infiltrating inflamed UC and CD mucosa, respectively [89]. UC is generally suggested to involve T helper2 (Th2) cells, whereas CD involves Th1 and/or Th17 cells. In the past ten years, VIP has emerged as a therapeutic candidate to treat inflammatory diseases involving Th1 components and both VIP and VIPR systems [90]. Wu et al. demonstrated the role of VIP in the development and maintenance of epithelial barrier integrity by promoting epithelial cell repair and homeostasis in DSS-induced colitis mouse model [91]. Immune dysfunction is a critical component of IBD pathogenesis and VIP has a very important role in IBD pathogenesis as it modulates immune activities by maintaining the expression level of IL-10 in regulatory B cells in the intestine that stabilizes homeostasis of immune function [92].

### 4.3. Natural Polysaccharides

Generally, natural polysaccharides such as cellulose, dextran, pectin, and chitosan have been used as drug delivery systems for colon because of their ease to work, nontoxic nature, and approval by the USFDA. The use of natural polysaccharides as delivery system also prevents premature release of drug in the small intestine and stomach and favors selective degradation in colon. Modified apple polysaccharide inhibits colitis by decreasing the level of IL-22 and by increasing the expression of IL-22BP [93]. Additionally, polysaccharide-rich extracts of *Eucheuma cottonii* and *Acmella oleracea* modulate inflammatory response and suppress colonic damage in DSS-induced colitis [94,95]. Nie et al. used non-starch polysaccharides to treat IBD in both in vivo and in vitro models [96]. The mechanisms involved in amelioration of the signs and symptoms and in suppression of reoccurrence rates are anti-inflammation, immune stimulation, and gut-microbiota modulation.

### 4.4. Bacterially Derived Immunomodulants

Beneficial bacteria (Probiotics) have therapeutic use in IBD treatment. However, only modest benefit has been reported so far in humans [97]. Enteric bacterial pathogens have coevolved with humans to develop systems to modulate inflammatory and immunoregulatory pathways [98]. AvrA (naturally evolved immunomodulatory protein), a member of acetyltransferases family, is produced by Salmonella that fits in this category. AvrA performs its function by covalently modifying and inactivating mitogen-activated protein kinase (MAPK), thus affecting growth, survival, and immune pathways in eukaryotes. Previous findings documented that overexpression of AvrA in transfected cells block activation of JNK (c-Jun *N*-terminal kinase), MAPK, NF-κB, and a range of inflammatory effector genes at the transcriptional level. Estrada et al. used engineered AvrA to suppress inflammatory response similar to those observed in IBD [99]. Further, cross linking of AvrA to nanoparticles (AvrA-NPs) makes their internalization possible into epithelial and lamina porpria monocytic cells in both in vitro and in vivo models. AvrA-NPs inhibit inflammatory pathways and reduce inflammation of tissues in murine models of colitis, thereby making bacterial protein-NP platforms effective therapeutics to fight chronic IBD [99].

With the recent development of 16s ribosomal RNA sequencing, extensive efforts have now been made to identify various microbiota and antigenic variation of the microbiota at the inflamed site and to design a microbiome-targeted NP delivery system. Yan et al. demonstrated use of a pectin/zein hydrogel bead system to deliver p40, a probiotic bacteria-derived soluble protein, to the mouse colon. The protein p40 activates EGFR (Epidermal Growth Factor Receptor) in colon epithelial cells, which in turn activates Akt and thereby promotes inhibition of apoptosis induced by inflammatory cytokines in both in vitro and ex vivo models. Reduction of apoptosis of colon epithelial cells alters barrier function and thus helps in treating intestinal injury during acute colitis induced by DSS [100].

### 4.5. Insect-Derived Bioactive Components

Insect-derived bioactive compounds such as *Bombyx mori* haemocyte, *Gryllus bimaculatus* extract, *Tetragonula carbonaria* extract, *Nasonia vitripennis* venom, glycosaminoglycan, cecropin A, silk fibroin, SibaCec, Cecropin-TY1, *N*-acetyldopamine dimers, papiliocin, and Melittin have been reported to be active against inflammatory diseases. These compounds inhibit the secretion of cytokines and downregulate expression of signaling molecules involved in the pathophysiology of inflammation [101,102]. They inhibit activation of MAPK pathways and NF-κB, thereby acting against inflammation under both in vitro and in vivo conditions [103]. However, the knowledge behind their regulatory activities is not sufficient. Further studies related to their ability to cross the blood–brain barrier along with toxicological and pharmacodynamical properties will direct them to be used as potential drugs for future clinical trials in colitis and CAC [103].

### 4.6. Engineered Biomimetic Nanovesicle

In IBD, a subset of T-lymphocytes overexpresses α4β7 integrin, which helps in binding to its receptor on the endothelial membrane. Based on this principle, biomimetic vesicles termed leukosomes (SLKs), which are leukocyte-like carriers drugged with over-induced α4β7 integrin, are engineered. In DSS-induced colitis model, SLK treatment reduces inflammation via affecting both inflammation-favoring and anti-inflammatory genes. SLKs also suppress infiltration of immune cells, which in turn enhances intestinal repair. Therefore, biomimetic nanovesicles serve both as natural drug delivery systems as well as nanotherapeutics with inherent anti-inflammatory properties [104].

### 4.7. Antibodies and Nucleic Acids

#### 4.7.1. CD98

CD98 is chosen in the studies conducted by Xiao et al. as the therapeutic targeting molecule [105]. Earlier studies reported upregulated expression of CD98 in mice colonic tissues suffering from UC [106]; in intestinal B cells, CD4^+^ T cells, and CD8^+^ T cells from IBD patients [107]; and in colonic biopsies from CD patients [108]. CD98 is reported to be redistributed to the apical surface of intestinal epithelium during inflammation, resulting in loss of epithelial barrier [109,110]. Also reported are the findings that increased expression of CD98 has an important role to play in the progression and development of IBD [111]. Administration of chitosan/alginate hydrogel nanoparticles linked with CD98 antibody and loaded with siRNA CD98 significantly reduces expression of CD98 in epithelial cells and macrophages. These NPs decrease severity of colitis in mice, suggesting future use of chitosan/alginate nanoparticle for IBD therapy [105]. Zhang et al. used natural nanoparticles known as ginger-derived lipid vehicles (GDLVs) produced from lipids found in ginger for the delivery of siRNAs. Oral administration of GDLVs loaded with a very low dose of siRNA-CD98 targets them specifically and effectively to colon tissues, which results in decreasing expression of CD98. Moreover, GDLVs are biocompatible and production on a large scale can be achieved easily, thus making them a very safe and cost-effective delivery system for siRNA in UC treatment [7].

#### 4.7.2. TNF-α

TNF-α is amongst the key genes involved in IBD pathogenesis, and many therapies involving use of antibodies for reducing the activity of TNF-α have been tested in several clinical trials; however, systemic depletion of TNF-α results in adverse effects [112,113,114]. Gene silencing with targeted delivery of TNF-α siRNA encapsulated in thioketal nanoparticles (TKNs) with B gelatin enclosed in poly ε-caprolactone (PCL) microspheres decreases TNF-α mRNA at the site of intestinal inflammation in the DSS-induced mouse model [115,116]. Other studies reported that the TNF-α siRNA/polyethyleneimine (PEI) nanocomplex inhibits secretion of TNF-α by macrophages whereas its oral administration in lipopolysaccharide (LPS)-treated mouse model reduces expression of TNF-α, specifically in colon [117].

Biodegradable poly(DL-lactide-co-glycolide) nanoparticles (PLGA-NPs) loaded with nucleic acids, for example, chitosan-modified (CS)-PLGA loaded with NF-κB decoy oligonucleotide (ODN), are shown to be useful in treating mice with colitis [118]. Similarly, oral administration of double-stranded decoy ODNs enclosed in CS-PLGA nanospheres (NSs) against the pro-inflammatory NF-κB gene has curative effects on diarrhea, blood loss, colon length, and MPO activity in the DSS-induced murine UC model [119].

#### 4.7.3. MAPK4

Aouadi et al. revealed that mitogen-activated protein kinase 4 (MAPK4) plays an important role in mediating the production of inflammatory cytokines in macrophages. Gene silencing by administration of MAPK4 siRNA encapsulated in β1,3-d-glucan shells in LPS-induced mouse model suppresses the production of TNF-α and IL-1β, thereby protecting the animals from systemic inflammation induced by LPS [120].

#### 4.7.4. MMPs

The architecture of intestinal tissue is disturbed in IBD during inflammation and wound healing. Matrix metalloproteinases (MMPs) play an important role in regulating tissue remodeling during these processes [121]. In an DSS-induced mouse model of UC, expression of MMP-3 (stromelysin-1) and MMP-10 (stromelysin-2) increased in gut and intestinal ulcer tissues [122]. Silencing of MMP-3 and MMP-10 gene by their respective siRNA has been documented as therapeutically beneficial in protecting damaged colonic tissues and severity of the disease [123].

#### 4.7.5. CyD1

Increased expression of Cyclin D1 (CyD1) is reported in epithelial and immune cells during IBD [124]. siRNA CyD1 linked to targeted stabilized NPs (tsNPs) directed against leukocytes inhibits CyD1 mRNA and inflammatory responses in DSS-induced mouse model of colitis. CyD1 silencing affects induction of inflammatory cytokines TNF-α and IL-12 in TH1 cells; however, no effect on IL-10 cytokine in TH2 cells is observed [125].

#### 4.7.6. IL-10

Anti-inflammatory IL-10 is very critical for immunosuppression in CD pathogenesis but therapeutic efforts to enhance its action have failed so far because of systemic toxicity of IL-10 therapies and reduced delivery of IL-10 to the intestinal tissues [126]. In a TNBS-induced acute colitis mouse model, targeted delivery of IL-10-producing plasmid in the form of pORF5-mIL-10 plasmic DNA with type B gelatin nanoparticles in poly ε-caprolactone resulted in enhanced expression of IL-10 followed by suppression of pro-inflammatory cytokines (TNF-α, IFN-γ, IL-1α, IL-1β, and IL-12), thus restoring body weight, length of colon, and beneficial clinical activity scores in animal models [127]. Lipid nanoparticles (LNPs) are natural, efficient, safe, and non-immunogenic systems for gene manipulation in vivo. LNPs shield RNA molecules from degradation and from eliciting any immune reaction. LNPs play a significant role in achieving therapeutic concentration of IL10 in colons while reducing the off-target protein expression [128]. As a therapeutic approach to treat IBD, Veiga et al. described the use of IL10-modified mRNA encapsulated in LNPs for cell-specific delivery into Ly6c^+^ inflammatory leukocytes in both in vitro and in vivo models [129].

#### 4.7.7. IRF-8

Inhibition of Interferon Regulatory Factor-8 (IRF-8), an immunomodulatory protein, has therapeutic potential in IBD. siRNA-loaded lipid-based nanoparticles (siLNPs) block IRF-8 mRNA and significantly regulate differentiation, polarization, and activation of mononuclear phagocytic cells. To silence IRF8 in vivo, siLNPs are coated with anti-Ly6C antibodies to achieve selectivity for inflammatory leukocytes. The immunomodulatory effect is observed with a significant decrease in pro-inflammatory cytokines [130].

#### 4.7.8. miR-29, miR-31, and miR-146b

Fukata et al. established a microRNA (miR)-based therapy in a DSS-induced colitis model. They used miR-29 and supercarbonate apatite (sCA) nanoparticle (sCA-miR-29) as the drug delivery system. Both miR-29a and miR-29b prevent inflammation when injected in the tail of a murine colitis model. RNA seq analysis revealed that inhibition of inflammatory cascade associated with interferon is responsible for the prevention of inflammation. Additionally, sCA-miR-29b when injected subcutaneously also inhibits inflammation by targeting CD11c^+^ dendritic immune cells of inflamed mucosa and by suppressing production of IL-6, TGF-β, and IL-23 subunits, thus suggesting sCA-miR-29 as a new route in nucleic acid-based medicine for IBD treatment [131].

Levels of miR-31 are increased in Crohn’s, colitis, and CAC patients. However, knockout of miR-31 in mouse models developed severe colitis by DSS and TNBS. Analyses revealed that miR-31 regulates IL-17, IL-7, and GP-130 (a cytokine signaling protein) mRNAs, which are upregulated in miR-31 knockout mice. Delivery of miR-31 or oxidized konjac glucomannan microspheres (OKGM)-peptosome miR-31 mimic inhibits their expression; suppresses inflammatory response; and augments epithelial cell proliferation, body weight, and colon length [132].

MiR-146b mimic on mannose-modified trimethyl chitosan (MTC)-conjugated nanoparticles (MTC-miR146b) selectively targets intestinal macrophages for mucosal regeneration. MiR-146b strongly inhibits M1 macrophage activation via the toll-like receptor 4 signaling pathway that results in the repression of the induction of TNF-α, IL-6, and IL-1β [133].

Summarized list of Phytochemicals/Macromolecules used in nano-based IBD research is included in Table 1. This table has some additional references which are not included in the text.

## 5. Future Challenges in the Use of NaturalProduct-Based Nanomedicine in IBD

### 5.1. Novel Unproven Anti-Inflammatory Nano-Formulations and Patents in Colitis

More recently, a new terminology “nutraceutical” has emerged, which is described as “A food or parts of food that provide medical or health benefits, including the prevention and/or treatment of disease”. This term originated from two words: “nutrition” and “pharmaceutical”. A recent publication summarized various novel nano-formulations and several patents, which include curcumin, resveratrol, quercetin, epigallocattchin-3-gallate, β-carotene, fish oil, and gallic acid. These novel formulations, which have anti-inflammatory and antioxidant activities, are fully characterized by efficacy and toxicity in laboratory models [134]. However, these formulations have never been investigated in colitis research. Future investigations using these novel nano-formulations and patents are warranted.

### 5.2. Combinatorial Nano-Formulations in IBD Treatment

Combination therapy, an emerging strategy, is under intensive preclinical investigation for the treatment of IBD. In recent years, nutritional supplements and natural macromolecules showed beneficial effects in IBD treatment either alone or in combination with other molecules or biologics. In this regard, zinc oxide nanoparticles combined with mesalazine (5-ASA) enhance the therapeutic efficacy of 5-ASA in the treatment of colitis [135]. Simultaneous delivery of CD98 siRNA in combination with curcumin using hyaluronic acid (HA)-functionalized polymeric nanoparticles is an effective technology to target cells for colitis therapy [136]. Nanoparticles which are pH sensitive are used to deliver curcumin-celecoxib combination as a potential therapy for colitis [137]. Coadministration of silymarin and nano-selenium inhibits NF-κB in the management of IBD [74]. Combination therapy of migraine with ω-3 fatty acids and nano-curcumin should also be tested in IBD [138]. Similarly, ω-3 polyunsaturated fatty acids loaded in resveratrol-based solid lipid nanoparticles, which suppresses CRC in vitro models, may also reduce IBD complication [139].

### 5.3. Natural Nanoparticles as Delivery Platforms in IBD Treatment

Phytochemicals can act on various targets of pathogenesis and inflammation. Owing to immense pharmacological activities, at present, the key limitations and the barriers observed with natural molecules in IBD therapy are that the therapeutic activity is compromised before reaching to the inflamed colon. To overcome these challenges, various synthetic polymer-based nanoparticles are being utilized that are toxic to biological systems. To minimize the risk of side effects associated with synthetic polymer-based nanocarriers, future studies should be aimed at delivering and/or targeting these molecules by encapsulating them into natural protein and/or polysaccharide-based delivery systems. Natural biopolymer-based nanoparticulate formulations can protect active phytochemical from gastrointestinal instability, can maximize colon targeting, and can get degraded by colonic microflora without altering their composition; hence, such a biodegradable carrier system can be an ideal formulation for colon targeting in the future. In this regard, a natural-lipid nanoparticle drug delivery system is created to encapsulate and release 6-shogaol (biologically active compound of ginger) to the colon [140]. *Ulva lactuca* polysaccharide-selenium nanoparticles offer therapeutic potential for reducing the symptoms of acute colitis through its anti-inflammatory actions [141]. Nanoparticles isolated from broccoli extracts provide protection against colitis by activation of adenosine monophosphate-activated protein kinase in dendritic cells [142]. Budesonide-entrapped krill oil-incorporated liposomes suppressed TNF-α in colitis models. This natural delivery platform has great potential as a nanovehicle for oral delivery of IBD drugs [143]. An exhaustive discussion on the different categories of nanocarriers is beyond the scope of this article.

### 5.4. Nanoparticles as Delivery Platforms in IBD Clinical Trials

Clinical literature suggests that molecules like curcumin, resveratrol, rutin, and silymarin are being investigated in humans for treatment of IBD or CRC. Two randomized double-blind placebo-controlled pilot studies established anti-inflammatory effects of resveratrol in colitis patients [155,156]. A review article establishes that various clinical trials with resveratrol caused improvement in clinical symptoms, endoscopic and histological assessment, and quality of life and reduction in adverse events [157]. In a recent double-blind placebo-controlled trial, curcumin supplementation in colitis patients is associated with significant improvement of the clinical outcomes and quality of life [158]. A recent review focused on the clinical trials that assessed herbal medicinal plants in double-blind randomized controlled trials (RCTs) [159]. In general, trials were conducted with single small RCTs and short follow-up. Further, long-term effects and safety of their use are not yet established. Thus, appropriately sized RCTs are important prior to recommended use of herbal medicines in therapy.

In the clinical trials discussed above, natural molecules are being used as such or as conventional oral dosages but not as a nanoparticulate formulation. This warrants future trials encompassing these phytochemicals into nontoxic nanoparticles to unlock their therapeutic potential for successful clinical outcome in IBD and IBD-CRC. In this regard, it was established that PEG-functionalized nanoparticles selectively target inflamed mucosa in IBD [160]. This could be further exploited to deliver desired phytochemicals in the treatment of IBD.

## 6. Conclusions

IBD therapy with biologics is favored for mucosal healing and for maintaining clinical remission. Invariably, biologics lead to adverse side effects, a key limitation observed in treatment of IBD. This review demonstrates that, as an alternate, natural products when entrapped in nanoparticles hold a huge potential in the prophylaxis, management, and treatment of IBD. In future, new untested patents and novel nano-formulations should be investigated. Further, combinatorial mixtures of phytochemicals or phytochemicals with macromolecules may have additive effects as compared to single agent treatment strategy. More importantly, though limited plant-based nano-formulations have been tested in humans, new clinical trials are urgently needed to analyse nontoxic natural nanoparticles targeted to colon for delivery of natural products.

## Figures and Tables

**Table 1 ijms-21-03956-t001:** Summarized list of Phytochemicals/Macromolecules used in nano-based Inflammatory Bowel Disease (IBD) research.

Phytochemical/Macromolecule	Nanoparticles/Carrier	In Vitro/In Vivo Models	Reference
Thymoquinone	LPS, alginate microcapsule	HT-29	[38]
Resveratrol	CTS-NPs	in vitro	[40]
Curcumin	CTS NPs	DSS/CAC	[54]
Curcumin	HA-PLGA-NPs	HT-29	[144]
Curcumin	C-SBLNPs	DSS	[145]
Curcumin	Theracurmin-HP	DSS	[146]
Curcumin	PEG	DSS	[53]
Curcumin	CTS NPs	DSS-AOM	[54]
Curcumin	SNEDDS, NLC, and LCSPNC	DSS	[1]
Ginger active compound, 6-shogaol	PLGA/PLA-PEG-FA and PLGA/PLA-PEG-FA-chitosan alginate hydrogel	colon-26, Raw 264.7, and DSS	[147]
Ginger	GDNPs 2	RAW 264.7, Caco-2BBE, Colon-26, and AOM-DSS	[13]
Quercetin	PEG-coated vesicles with CTS	TNBS	[58]
Quercetin, Glycoside	Glycoside-Rutin	DSS	[59,60]
Embelin	Cellulose and Eudragit S 100 polymer microspheres	AA and DSS	[68]
Grape Exosome	GELNPs and LLNs	DSS	[72,73]
Silymarin	Selenium NPs and Eudragit NPs	TNBS and AA	[74,75]
Caffeic acid phenethyl ester (CAPE)	Albumin	DSS	[82]
Piceatannol (PCT)	Albumin	DSS	[82]
Tuftsin	Phosphocholine	DSS	[85,86]
Vasoactive Intestinal Peptide	SSM NPs	DSS	[88,91]
**Bacteria**			
Salmonella acetyltransferase, AvrA	eGFP cross linked DTSSP NPs	in vitro, DSS	[99]
Probiotic derived protein p40	Pectin/Zein Hydrogel	in vitro, ex vivo, and DSS	[100]
Colonic bacteria	ZnO NPs	DSS	[135]
**Engineered Biomimetic Nanovesicles**			
α4β7 integrin	SLKs	DSS	[104]
**Antibodies**			
Hsp60	Virus-like particles	DSS	[148]
Ly6C	Lipid-based NPs	DSS	[130]
Mucosal addressin cell adhesion molecule-1	MnO NPs	DSS	[149]
scCD98	CTS and alginate hydrogel	DSS	[105]
Tumor necrosis factor-alpha	*Lactococcus lactis* nanobodies	DSS	[150]
**Nucleic Acids**			
NF-kB decoy oligonucleotide	CTS-PLGA NSs	DSS	[118]
Plasmid DNA containing PIAS1	TAC6 polymer-sodium polyaspartate NG	DSS	[151]
IL-10 containing plasmid	Type B gelatin NPs-PCL	TNBS	[127]
IL-10 RNA	LNPs	in vivo	[128]
IL-10 Modified mRNA	LNPs	RAW 264.7 DSS	[129]
CD98 siRNA	GDLVs	Caco-2BBE, RAW 264.7, colon-26, and DSS	[7]
CD98 siRNA	NPs and NPs-chitosan/alginate hydrogel	Colon-26, RAW 264.7, and DSS	[105]
CyD1 siRNA	tsNPs	DSS	[125]
TNF-α siRNA	GC-NPs	DSS	[152]
TNF-α siRNA	TKNs-B Gelatin-PCL	DSS	[115]
TNF-α siRNA	TPP-PPM NPs	DSS	[153]
TNF-α siRNA	PLA-PEG-chitosan/alginate hydrogel	DSS	[154]
TNFα siRNA	PEI/PL/PVA	MPs and LPS	[117]
MAPK4 siRNA	β1,3-d-glucan shells	LPS	[120]
miR31	OKGM	DSS	[132]
miR-29	sCA NPs	DSS	[131]

LPS, Lipopolysaccharide; CTS-NPs, Chitosan-based biocompatible hydrogel nanoparticles; CTS, Chitosan; HA-PLGA, Hyaluronan-Poly(lactide-co-glycolide)acid; cSBLNPs, Solid binarylipid nanoparticles; SNEDDS, Self-nanoemulsifying drug delivery systems; NLC, Nanostructured lipid carriers; LCSPNC, lipid core-shell protamine nanocapsules; GDNPs, Ginger-derived nanoparticles; PEG, Polyethylene glycol; GELNPs, Grape exosome-like NPs; LLNS, liposome-like nanospheres; SSM, Sterically stabilized micelles; LNPs, lipid NPs; GC, Galactosylated chitosan; PLA-PEG, Poly(lactic acid)Poly(ethylene glycol); TPP-PPM NPs, Sodium triphosphate-mannosylated bioreducible cationic polymer NPs; PEI-PL-PVA, Polyethyleneimine-polylactide-polyvinylalcohol; OKGM, Oxidized konjac glucomannan microspheres; sCA, Supercarbonate apatite NPs; TKNs, Thioketal; PCL, poly (ε-caprolactone); PIAS1, protein inhibitor of activated STAT1; tsNPs, Targeted stabilized NPs; DTSSP, dithiobis(sulfosuccinimidylpropionate).

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
