# Peer review of "Natural Product-Based Nanomedicine in Treatment of Inflammatory Bowel Disease"

_ijms, 2020, doi:10.3390/ijms21113956_

Round 1

Reviewer 1 Report

My only 3 comments are:

  1. Understanding that the authors did not embark on a systematic review, then how did the authors decide on these set of natural phyotchemicals/macromolecules? There's no description. If so why did they choose this list and not any other? And which years and how extensive was their review process. If there's limited evidence could the authors consider a systematic review instead so that the level of this evidence improves and therefore increases the chances of it being accepted for publication?
  2. The introduction and / or conclusion can describe the barriers / future considerations for the utility of these compounds for IBD, this description would add weight and clinical / research utility of this review. In the conclusion, the authors merely only state " In future studies, mixture of phytocohecmicals from different pant based formulations could also be explored." There is no explanation. 
  3. The title should just read: Natural Product-Based Nanomedicine in Treatment of chronic Inflammatory colitis and the associated cancer pathway. Or consider just stating IBD.

Otherwise, generally informative and a summary of some evidence, do consider the above points for relevance to readership.

Author Response

We appreciate the critical comments provided by the reviewers. As per comments from all the reviewers, we have added references searched in PubMed with additional search criteria, which is now clearly stated in the last paragraph of Introduction. Further, in the introduction we have also introduced current treatments of IBD. Translational potential is also discussed in revised manuscript section-5. Future, studies with novel formulations, combinatorial formulations and patents, significance of natural nanoparticles in delivery and clinical trials is discussed in section-5.

  1. Understanding that the authors did not embark on a systematic review, then how did the authors decide on these set of natural phyotchemicals/macromolecules? There's no description. If so why did they choose this list and not any other? And which years and how extensive was their review process. If there's limited evidence could the authors consider a systematic review instead so that the level of this evidence improves and therefore increases the chances of it being accepted for publication?

Response: In last section of introduction, we have now stated search terms used in PubMed and the search was made for the last ten years. In this process we obtained few additional references due to addition of new terms. It is pertinent to mention here that we have mostly included studies where nanoparticles are used to entrap natural products and also studies with macromolecules. In some studies macromolecules also serve as nanoparticles. In the new section-5, we have mentioned that this is not an exhaustive review on nanoparticles as such. However, we have discussed the importance of natural nanoparticles for future clinical trials.

  1. The introduction and / or conclusion can describe the barriers / future considerations for the utility of these compounds for IBD, this description would add weight and clinical / research utility of this review. In the conclusion, the authors merely only state "In future studies, mixture of phytocohecmicals from different pant based formulations could also be explored." There is no explanation.

Response: As per suggestion, we have updated introduction and conclusion. We have discussed translational challenges and utility of natural products in terms of novel nanoformulations and clinical trials in section-5. Key limitations and barriers also included inside the manuscript mostly in Introduction and section-5.

  1. The title should just read: Natural Product-Based Nanomedicine in Treatment of chronic inflammatory colitis and the associated cancer pathway. Or consider just stating IBD.

Response: As suggested by reviewer, authors have changed the manuscript title as “Natural Product-Based Nanomedicine in Treatment of Inflammatory Bowel Disease”.

We believe that thoughtful suggestions provided by the esteem reviewer has considerably improved this review article. We sincerely hope that major additions will satisfy the queries made by the reviewer.

Reviewer 2 Report

Authors provided an overview of several nanoparticle formulations of natural compounds for their potential role in IBD treatment.

The review is quite clear and interesting, as nutraceuticals have been heavily investigated in recent years. In the introduction section, the references are scarce despite the wide literature on the potential role of natural phytochemicals in IBD therapy (see the recent Larussa T et al., Int J Mol Sci. 2019 Mar 20;20(6). pii: E1390; Dicarlo M et al., Int J Mol Sci. 2019 Nov 16;20(22). pii: E5771 and many others)

In the conclusion section, Authors mention biologics but no comments are present in the Introduction section regarding anti-TNFalpha and other biologic drugs currently used in the IBD therapy. Authors should shortly mention this category of drugs in the Introduction section, as they represent the most recent and innovative instrument for IBD management but at the same time their cost and safety arise concerns in clinicians.

Table 1 is carefully elaborated. English language is fine.

Author Response

We appreciate the critical comments provided by the reviewers. As per comments from all the reviewers, we have added references searched in PubMed with additional search criteria, which is now clearly stated in the last paragraph of Introduction. Further, in the introduction we have also introduced current treatments of IBD. Translational potential is also discussed in revised manuscript section-5. Future, studies with novel formulations, combinatorial formulations and patents, significance of natural nanoparticles in delivery and clinical trials is discussed in section-5.

  1. The review is quite clear and interesting, as nutraceuticals have been heavily investigated in recent years. In the introduction section, the references are scarce despite the wide literature on the potential role of natural phytochemicals in IBD therapy (see the recent Larussa T et al., Int J Mol Sci. 2019 Mar 20;20(6). pii: E1390; Dicarlo M et al., Int J Mol Sci. 2019 Nov 16;20(22). pii: E5771 and many others).

Response: Authors have included additional latest information regarding natural phytochemicals in the introduction and in relevant subsections as per suggestion. Both references mentioned above are included in the highlighted sections. All together, we have added 33 new references. In the new section-5, we have mentioned that this is not an exhaustive review on nanoparticles as such. However, we have mostly included studies where natural products/phytochemicals entrapped in nanoparticles are investigated.

  1. In the conclusion section, Authors mention biologics but no comments are present in the Introduction section regarding anti-TNFalpha and other biologic drugs currently used in the IBD therapy. Authors should shortly mention this category of drugs in the Introduction section, as they represent the most recent and innovative instrument for IBD management but at the same time their cost and safety arise concerns in clinicians.

Response: We completely agree with the suggestion and therefore necessary details have been incorporated in the introduction. Apart from this, we have also created a new section 5 for future research in this area. Additionally, limitations and clinical implications are also included to increase the significance of this review.

We believe that thoughtful suggestions provided by the esteem reviewer has considerably improved this review article. We sincerely hope that major additions will satisfy the queries made by the reviewer.